# Development of a Nomogram to Estimate the 60-Day Probability of Death or Culling Due to Severe Clinical Mastitis in Dairy Cows at First Veterinary Clinical Evaluation

**DOI:** 10.3390/vetsci10040268

**Published:** 2023-04-01

**Authors:** Thomas Le Page, Sébastien Buczinski, Jocelyn Dubuc, Josiane Labonté, Jean-Philippe Roy

**Affiliations:** Faculté de Médecine Vétérinaire, Université de Montréal, Montreal, QC H3T 1J4, Canada

**Keywords:** severe clinical mastitis, dairy cattle, nomogram, predictive model, prognosis

## Abstract

**Simple Summary:**

Severe clinical mastitis is an inflammation of the mammary gland with systemic clinical signs. It is a frequent and highly fatal disease of dairy cattle worldwide. Nomograms are practical tools that help predict a specific issue based on statistical models. Here, we developed nomograms for prediction of death or culling related with cases of severe clinical mastitis that were treated in dairy farm settings following a veterinary clinical examination. Such a tool will help veterinarians in prognosis prediction. A reliable prognosis is important to make euthanasia decisions in poor prognosis cases.

**Abstract:**

Severe clinical mastitis is a frequent disease of dairy cattle. An effective mean of predicting survival despite treatment would be helpful for making euthanasia decisions in poor prognosis cases. The objective was to develop a nomogram for prediction of death or culling in the 60 days following a severe mastitis episode in dairy cows at first veterinary visit in farm settings. A total of 224 dairy cows presenting severe clinical mastitis and examined for the first time by a veterinarian were included in a prospective study. Clinical and laboratory (complete blood cell count, L-lactate, cardiac troponin I, milk culture) variables were recorded. Animals were followed for 60 days. A nomogram was built with an adaptive elastic-net Cox proportional hazards model. Performances and relevance were evaluated by area under the receiver operating characteristic curve (AUC), Harrell’s concordance index (C-index), calibration curve, decision curve analysis (DCA) and misclassification cost term (MCT). The nomogram included: lactation number, recumbency, depression intensity, capillary refilling time, ruminal motility rate, dehydration level, lactates concentration, hematocrit, band neutrophils count, monocyte count, and milk bacteriology. The AUC and C-index showed a good calibration and ability to discriminate. The DCA suggested that the nomogram was clinically relevant. Euthanizing animals having less than 25% probability of survival is economically optimal. It could be used for early euthanasia decisions in animals that would not survive despite treatment. To facilitate the use of this nomogram by veterinarians, a web-based app was developed.

## 1. Introduction

Severe clinical mastitis in cattle is defined as a modification of mammary secretions with signs of quarter inflammation and at least one of the following systemic clinical signs: increased body temperature, dehydration, ruminal hypomotility, recumbency, or signs of depression [1]. The reported incidence of this disease is 1.8–15.3 cases per 100 cow-years [2,3,4]. Mastitis can be caused by many different pathogens (bacteria, yeast, or algae) [5]. Some of these pathogens are more frequently associated with episodes of severe clinical mastitis. Of those, coliforms (i.e., *Escherichia coli* (*E. coli*) and *Klebsiella* spp.) are related to poor outcomes [4]. *Staphylococcus aureus* (*S. aureus*) is also related to fatal episodes of severe clinical mastitis called gangrenous mastitis [6]. In some countries (e.g., Germany), *Streptococcus uberis* (*Str. uberis*) is an increasingly frequent pathogen in severe clinical mastitis [7]. Weinz et al. reported a high fatality and culling risk in the weeks following an episode of severe clinical mastitis (19% and 52%, respectively) [1], while other authors have reported fatality risks of 11% and 13.5% in the weeks following an episode [4,8]. Because of the negative impact of severe mastitis in cows, survival of these animals remains a concern for veterinarians and dairy producers. It would be helpful for veterinarians and dairy producers to estimate this condition survival probability properly.

In veterinary medicine, early prediction of a fatal issue offers the opportunity of an early euthanasia to avoid unnecessary suffering to the animal. This prediction could also improve the farmer’s availability for animal care by reducing time spent treating an animal with a low probability of recovery and allowing more time for taking care of the other animals in the herd. It could also reduce potential farmer psychological distress related to the deterioration of a cow despite treatment and time devoted to her. Finally, it could also become economically relevant by reducing the cost of a useless treatment and related manpower costs.

Several tools could be used to estimate the survival probability of an animal affected by severe clinical mastitis. These are generally related to the detection of sepsis or endotoxemia (e.g., clinical examination or complete blood count (CBC) for detection of signs of severe inflammatory response syndrome [9]). While they have a general medical predictive ability for survival, none of them have been specifically evaluated in the case of severe clinical mastitis. L-lactate and cardiac troponin I (cTnI) have been studied as tools to establish a prognosis for different diseases of cattle [10,11,12]. Hypertroponemia is generally related to primary or secondary myocardial injuries [13], while hyperlactatemia generally reflects the hypoperfusion of an organ and potential sepsis [14,15]. Interestingly, the association between troponemia or lactatemia and subsequent survival of cows affected by severe clinical mastitis has not been reported. Both parameters could potentially be useful to establish a prognosis and deserve investigation.

An interesting methodology to quantify the probability of a specific outcome is the use of predictive scores. They are a way of obtaining the probability of a medical event based on different variables (clinical or paraclinical). They are extensively used in human medicine to make evidence-based predictions and allow appropriate treatment decisions [16,17,18]. Using predictive scores following the first clinical examination of an animal is novel in the field of veterinary medicine. To ensure appropriate inference and usefulness of this approach, it is important to perform a validation of predictive scores [17]. Penalized regression techniques are techniques that can be used to compute predictive models with good performance when applied in a different population to where the model was built [19].

Nomograms are graphical representations of predictive models [18]. To our knowledge, none have been developed for the prediction of a clinical issue in veterinary medicine. They are a means of easily determining the probability of a medical event (e.g., survival or disease) in the function of a set of predictors (e.g., age of an animal or level of a biochemical parameter). Two options are available to use a nomogram for a clinical decision. The first option is a representation of the nomogram on a graph. Each predictor is given a scale of points corresponding to its magnitude. The sum of all the points given to each predictor is then used to determine a predicted probability on another scale. An example is available in Appendix A and more information is provided by Balachandran et al. [18]. Web-based applications are the second option and are a more modern and precise way of using a nomogram in clinical settings.

The objective of this study was to develop a predictive model for death or culling caused by an episode of severe clinical mastitis in dairy cattle after a first veterinary clinical examination of the animal at the farm, and with that model to develop a nomogram to predict the death or culling of an animal in the next 60 days. A secondary objective was to compare two predictive models: the first one considered only predictors available after a veterinary clinical exam and the second one considered all clinical and laboratory predictors. Finally, the output of these two models were compared with the prognosis made by veterinarians based on clinical experience.

## 2. Materials and Methods

This study was developed following the Transparent Reporting of a multivariable prediction model for Individual Prognosis or Diagnosis (TRIPOD) checklist [16] (Appendix A). The study protocol was accepted by the ethical animal care use committee of the Faculté de médecine vétérinaire, Université de Montréal (project number Rech-1606). The first nomogram was built in six steps: 1. collection of data with a prospective cohort study; 2. descriptive analysis of the sample; 3. handling of missing data via multiple imputations; 4. development of a nomogram with a penalized regression technique; 5. internal validation of the statistical performances of the nomogram; and 6. assessment of the clinical and economical relevance of the nomogram. To answer the secondary objectives of the study, we developed a second nomogram with clinical predictors only following steps 4–6, and we compared the clinical relevance of the nomograms to the prognosis that veterinarians gave at first visit. For more details, a general description of nomogram building process and interpretation is described by Balachandran et al. [18].

### 2.1. Data Collection

The unit of interest was a cow affected by severe clinical mastitis. A prospective cohort study was conducted between July 2011 and September 2013, together with another study focusing on the survival of downer cows [10]. Initially, the purpose of that study was to evaluate L-lactate and cTnI measurements as prognosis factors for severe clinical mastitis. An estimated overall sample size of 200 cows affected by severe clinical mastitis was computed based on a difference of 20 percentage points between groups for high and low fatality risks (5% vs. 25%), as well as alpha and beta errors of 5% and 10%, respectively. A 10% loss to follow-up was also anticipated. Therefore, a total of 220 dairy cows affected by severe clinical mastitis were targeted for enrollment.

Five dairy veterinary practices located in the province of Québec, Canada, were involved in data collection. Cows were enrolled by veterinarians when detected ill by their owner. Inclusion criteria were: (1) at least one quarter with signs of inflammation (firmness, redness, edema, pain) and modified milk secretions (watery, blood, clots, pus); and (2) at least two of the following systemic signs: hyperthermia, anorexia, dehydration, or general weakness. If the inclusion criteria were met, a questionnaire with different information about the cow was completed: veterinarian identification (name and veterinary clinic); identification of the animal (herd, breed, number of days in milk (DIM), parity, previous mastitis history, and previous treatment for this case of mastitis); physical exam (recumbency, general clinical state (normal, slightly depressed, or severely depressed), appetite (from 0% to 100% by 25% steps as subjectively judged by the producer), rectal temperature, heart rate (beats per min.), respiratory rate (respiration per min.), ruminal motility rate (per min.), mucosal aspect (normal, congested, or pale), capillary refilling time (CRT) (<2, =2, >2 s), ruminal motility description (complete or incomplete), dehydration (<5%, 5–7%, 8–10% or >10%), feces appearance (diarrhea, blood, melena, fibrin, undigested, and/or, dry), quantity of feces (absence, small, or normal amount), udder exam (affected quarters, California mastitis test result, quarter evaluation (normal, slight firmness, moderate firmness, severe firmness, fibrotic tissue, or edema)), milk appearance (normal, presence of clots, purulent, aqueous or bloody secretions)); and administered treatments. The initial “best-guess” prognosis estimated by the treating veterinarian based on his or her experience at the first clinical examination was also recorded (poor, fair, good).

A blood sample was collected for a complete blood count (CBC), kept cool (4 °C) and sent to the Biovet laboratory (Saint-Hyacinthe, Québec, Canada). Parameters evaluated by an automate (Advia^®^, Siemens Healthcare, Erlangen, Germany) were packed cell volume or hematocrit (PCV, calculated), red blood cell count (RBC), hemoglobin concentration (HbC), mean globular volume (MGV), mean corpuscular hemoglobin (MCH, calculated), mean corpuscular hemoglobin concentration (MCHC), platelets, leucocytes, neutrophils, band neutrophils, lymphocytes, monocytes, eosinophils, and basophils. Fibrinogen was assessed with the heat precipitation technique. Presence of toxogram signs in neutrophils (Döhle body inclusion) was also assessed by a board-certified pathologist. L-lactate concentration (mmol/L, with a portable analyzer, Lactate Pro Arkray, Kyoto, Japan) and cTnI concentration (ng/mL, i-STAT analyzer, Abaxis, Union City, CA, USA) [13] were also quantified at the farm or veterinary practice.

A milk sample of the affected quarters was collected aseptically by the veterinarian [5]. A standard milk culture following National Mastitis Council guidelines [5] was realized at the diagnostic laboratory of the veterinary faculty (Centre de diagnostic vétérinaire of the Université de Montréal, Québec, Canada) or at the Biovet laboratory (Saint-Hyacinthe, Québec, Canada). Samples were considered positive for a specific pathogen when having 100 or more colony forming units per milliliter of milk. A milk sample from which three or more different species were cultured was considered contaminated. Contaminated results were considered as missing data.

The veterinarians were aware of the final CBC and milk culture results obtained after the initial farm visit but were blinded to L-lactate and cTnI results. Treatment management was performed by the veterinarians, without any supervision by the members of the research team. The initial treatment could potentially have been adapted based on the initial CBC and milk culture results.

The main outcomes of interest of this study were culling (animal withdrawn from the farm) and death (from clinical mastitis or another cause including euthanasia) of the enrolled cows. If one of the two happened, it was considered a negative outcome (“non-survival”). These outcomes were obtained via a phone follow-up interview with the animal owner using a standardized questionnaire performed on days 7, 30, and 60 after enrollment. There was no further investigation in case of a negative outcome. In case of loss of follow-up, cows were censored at the last follow-up available.

### 2.2. Descriptive Statistical Analyses

Data were originally collected using Access software (Microsoft, Redmond, WA, USA). A first analysis of the data was performed using Excel software (Microsoft, Redmond, WA, USA), for coding of missing data and renaming all variables. All other analyses were performed using R software (version 4.2.2; R Foundation for Statistical Computing, Vienna, Austria) [20].

Descriptive statistical analyses of the original dataset were computed with the *gtsummary* package [21]. Non-parametric univariable tests were computed for each variable (Pearson’s chi-squared test, Wilcoxon rank sum test, and Fisher’s exact test according to the nature of the variable).

### 2.3. Handling of Missing Data Using Multiple Imputation

Two cows having more than 50% of their data missing and a complete loss of follow-up were excluded. A high number of cows had at least one predictor missing (*n* = 191/222), but overall, the total amount of missing data was low (8.6%). After examination of missing data patterns, the data were considered missing at random (MAR) [22]. Imputing a missing value is generally preferred to deleting a case so that the maximum information is used and the bias that may result from a deleted case can be avoided [23].

Multiple imputation of the missing data was performed with the *mice* package following the process described by Van Buuren [23,24]. The entire dataset was used for the imputation. Firstly, a prediction matrix was defined using the function *quickpred*. Highly correlated variables (PCV, RBC, and HbC) were prevented from use for each other’s prediction. Veterinarian prognosis was predicted without the issue variables (time before event and event) and laboratory analysis (CBC, cTnl, and L-lactate). A visit sequence was defined going with increasing proportion of missing data through the predictors. Methods of imputation were defined for each type of variable (weighted predictive mean matching for numerical data, logistic regression for logical data, proportional odds model for ordinal data, and polytomous logistic regression for nominal data). Five imputed datasets were computed. Convergence was assessed graphically. Plausibility of the imputed data was checked with the density plot and strip plot function of the package.

### 2.4. Development of Survival Models Using a Penalized Regression Technique

Imputed data were vertically stacked (i.e., were assembled to form one dataset) to allow predictor selection methods. This method provides good results in low missing data ratios [25].

A first elimination of predictors was realized by the authors without any prior knowledge of the dataset. Three scientific and clinical criteria were used to eliminate a predictor from the dataset: high subjectivity of a predictor (mastitis history, mucosal aspect, milk and udder aspect, qualitative ruminal motility description), no prognosis interest of a predictor (breed, CMT score), and redundant predictors (hemoglobin and red cell count MGV, MCHC, and MCH) were removed in favor of the hematocrit. All other predictors were kept for model building.

Feces aspect was dichotomized into signs compatible with sepsis (diarrhea, blood, fibrin, and mucus) and others to limit the number of categories and the related over-fitting risk. Bacteriology was divided into negative culture, gram positive, gram negative bacteria, and yeast for simplicity of interpretation. In case of a mixed infection, animals were included in each category of the pathogens found.

A first nomogram was built with all the selected predictors using a penalized Cox proportional hazard regression model trained with an adaptive elastic-net procedure using 10-fold cross-validation on the whole sample for model selection. This method consists of the creation of a first model via an elastic-net regression, and then to use the coefficient found to weight the penalty of a second model (elastic-net). The penalty factor (lambda) and mix of L1 and L2 penalization (alpha) were chosen via cross-validation (minimal misclassification plus one standard error rule) in both instances. The main advantages of such a technique are: 1. the predictor selection ability for parsimony (especially in a situation where the number of predictors is higher than ten times the number of events, as in our sample); 2. the prevention of over-fitting; and 3. the possibility to use correlated predictors [19]. More information on the use of penalized regression techniques in predictive model building in the medical field are provided by Pavlou et al. [19]. The model was obtained with the *glmnet* package [26,27]. Finally, a web-based version of the nomogram was created with the *shiny* package [28]. It is available here: https://tlp-umontreal.shinyapps.io/paraclinical_nomogram/ (accessed on 29 March 2023). It provides an easy and didactic way of using our model for bovine practitioners. Linearity (Martingale residuals) and proportional hazard (Schoenfeld scaled residuals) assumptions were assessed graphically with the *survminer* package [29].

### 2.5. Internal Validation of the Models

C-index provides a simple way to assess the discriminative ability of a survival model. It is the proportion of pairs of cases where the subject with the higher predictive probability of non-survival has a shorter survival compared to the other. The C-index was obtained with the *glmnet* package [26,27]. For a survival model, area under the receiver-operator curve (AUC) is better represented as dependent of time of prognosis (time-dependent AUC or tAUC). It provides more precise information of discriminatory abilities over time. A calibration curve is a graphical way of assessing if the prognosis is reliable. The calibration curve and tAUC [30,31] were assessed on 100 bootstrapped samples with the *hdnom* package [32].

### 2.6. Assessment of the Clinical and Economical Relevance of the Models

Decision curve analysis plots the clinical benefit provided by the nomogram as a function of the concern of the user toward the intervention (here, toward euthanasia). It is a graphical way of assessing the clinical relevance of a test. Clinical benefit is defined as the number of true positives minus the number of false positives multiplied by a threshold factor [33]. This factor considers the sensitivity of the test-user toward a false positive (FP), in other words toward an excessive intervention (here, euthanasia of a cow that would have a positive outcome). The more extreme the impact of an excessive intervention, the more a false positive case is costly and thus, the less beneficial the test. Vickers et al. provided an extensive explanation of this concept [34]. Decision curve analysis (DCA) was computed with the *dcurves* package [33].

The misclassification cost term (MCT) is a metric of the relative cost of a false negative (FN) (non-survival of an animal that was not euthanized) compared to a false positive (FP) (euthanasia of an animal that would survive). Here FN costs would encompass administered treatments, time spent for treatment (veterinarian and farmer), normal care of the animal (food, bedding, …) minus the costs of euthanasia; FP costs would encompass the cost of a cow equivalent to the cow 60 days after mastitis and production during those 60 days minus the costs of administered treatments, time spent for treatment, normal care of the animal. The MCT is a graphical way of assessing the economical relevance of a test, it shows whether a test has a positive economic impact on a decision. It is also a useful tool to determine an economically optimal threshold for intervention. It was calculated as described by Buczinski et al. [35], with the predicted probabilities at day 60 (as in a binomial model). Censored data at day 60 were excluded from this calculation. We used the prevalence of non-survival in the sample for calculation of the MCT. As there is no formal evaluation of the specific cost of a severe clinical mastitis episode, a FN was deemed as less costly than a FP. We roughly estimated plausible ratios to be between 1 and 0.2. This variety of thresholds also reflects a high variation of the expected economic impact of a severe mastitis which depends on a lot of factors (time spent after the event, type of bacterium involved, production of the cow after the episode, etc.) The misclassification cost term was obtained from a code written by the authors. The whole code is available at https://github.com/tlpudem/nomograms (accessed on 29 March 2023).

### 2.7. Clinical Only Nomogram and Comparison with Veterinarian Prognosis

A second nomogram was built using the same approach. In this second model, only the predictors available with a clinical exam were used (exclusion of L-lactate, cTnI, CBC parameters, and bacteriology results). A web-based version of the nomograms was created with the *shiny* package [28]. It is available here: https://tlp-umontreal.shinyapps.io/clinical_nomogram/ (accessed on 29 March 2023).

To compare the nomograms with the veterinarian’s prognosis, the prognosis “Good”, “Fair”, and “Poor”, were transformed as a numerical probability of survival of 75%, 50%, and 25%, respectively. The DCA considering these probabilities was calculated for comparison of the clinical relevance of the prognosis and the models.

## 3. Results

### 3.1. Descriptive Results

A total of 224 cases of severe clinical mastitis from 124 different dairy herds were enrolled in the study. Three cows were enrolled twice after a two-month period (two cows with a different pathogen each time) and after one-year period (one cow). Two cases were excluded due to a complete loss of follow-up, and more than 50% of missing data. Most cases were recruited from one veterinary practice (*n* = 186/222; 84%). A total of 27 veterinarians were involved in the study, with a median number of 5.5 cases per veterinarian (min: 1, max: 34). Most cows were Holstein breed (*n* = 210/219; 96%). At enrollment, cows had a median of 30 DIM (Inter-quartile range (IQR): 3–90; missing data (MD) = 10), and the median number of lactations was three (IQR: 2–4; MD = 38). A total of 61% of the cows had a treatment started before the veterinary visit (*n* = 132/217). A total of 27% of the cows died within 60 days following the clinical mastitis episode (*n* = 54/209), of which 39% (21/54) were euthanized. All those deaths were attributed to the severe clinical mastitis episode. In total, 54% of cows were ousted from the farm during the same period (*n* = 112/209). Reasons invoked for culling an animal were: poor production (*n* = 25), mastitis or high somatic cell count (*n* = 12), lameness (*n* = 1) and unspecified reasons (*n* = 5). A total of 14 cases had a missing cause of culling,

Descriptive statistics of the clinical examination (overall and stratified by survival status) are presented in Table 1.

A table with descriptive statistics about prescribed and administered treatments by the veterinarians is available as Appendix A. Treatments included systemic antibiotics (*n* = 212/217), intramammary antibiotics (*n* = 186/213), non-steroidal anti-inflammatory drugs (NSAID; *n* = 206/218), intravenous fluids (*n* = 175/215), and calcium supplementation (*n* = 114/215). Descriptive statistics of blood laboratory analyses (CBC, cTnI, and L-lactate analyses) stratified by survival status are presented in Table 2.

There was a difference in bacteriology results between the two groups (survival vs. non-survival; *p* = 0.047, Fisher’s exact test). Descriptive statistics of milk bacteriology culture stratified by survival status are listed in Table 3.

### 3.2. Predictive Nomograms

#### 3.2.1. First Nomogram: Clinical and Laboratory Predictors

The nomogram with all the predictors is presented in Figure 1. It predicts the overall survival (no death or culling) in the 60 days following a first veterinary visit for a severe clinical mastitis case in dairy cows. Lactation number, recumbency, depression intensity, CRT, ruminal motility rate, dehydration status, L-lactate concentration, hematocrit, band neutrophil count, monocyte count, and milk bacteriology results were the retained predictors for this first nomogram. The assumption of proportional hazards was validated, as well as the linearity of the predictors.

The C-index of the nomogram was 0.71 ± 0.01, and the tAUC is presented in Figure 2. These two metrics show that the discriminatory abilities of the first nomogram were good (tAUC of 0.8) in the first 10 days after enrollment but decreased afterwards (tAUC of 0.65 around day 50). Overall, the discriminatory abilities of the nomogram were fair to good (C-index and tAUC). We can conclude that the nomogram is able to predict survival from non-survival better than chance.

The calibration curve of the nomogram at day 60 is presented in Figure 2. The calibration of the model was good: the calibration curve of the model is close to the straight line which represents a perfect calibration curve. In other words, these three metrics show that the model gives a reliable predicted probability (good calibration of the model). For example, 60% of the animals with 60% of predicted survival probabilities will survive.

Decision curve analysis of the nomogram evaluated at day 60 is presented in Figure 3. It shows that the model was beneficial in all investigated ranges (50% to 90% of probability of non-survival). This means that the model has a clinical benefit for all users who would decide to euthanize a cow; it includes the more interventionist users (euthanasia for all cows above 50% chances of non-survival) and the less interventionist ones (euthanasia for cows above 90% chances of non-survival).

The misclassification cost term of the nomogram at day 60 is presented in Figure 4. It plots the cost of errors of classifications—the higher, the costlier—in a function of the probability of non-survival above which the decision to euthanize is made. Firstly, the figure shows that a “euthanize none” policy (i.e., 100% is the probability of non-survival above which the decision to euthanize is made) is less costly than a “euthanize all” policy (i.e., 0% is the probability of non-survival above which the decision to euthanize is made) in nearly all scenarios (except with a ratio of cost FN/FP of 1). Secondly, the optimal scenario (the minimal cost) is between these two policies in all scenarios. The optimum threshold (the lowest point of a curve) varies between 55% (ratio FN/FP of 1) and 85% (ratio FN/FP of 0.2). The costlier a FP (euthanasia of a cow that would have a positive outcome) is compared to a FN (non-survival of an animal that was treated and not euthanized), the higher the probability of death at which euthanasia is decided should be.

#### 3.2.2. Second Nomogram: Clinical Predictors Only

The nomogram obtained with only predictors from the clinical exam is presented in Figure 5. Lactation number, being a downer cow, depression intensity, temperature, CRT, ruminal motility rate, and dehydration status were included in our second nomogram.

The assumption of proportional hazards was validated, as well as the linearity of the predictors. Time-dependent AUC and calibration curve of the second nomogram are presented in Figure 6. The second nomogram has slightly worse discriminative abilities compared to the first nomogram: tAUC of less than 0.8 at all times. The same tendency of loss of discriminatory capabilities over time is observed: tAUC is going downwards over time. Discriminatory capabilities tend towards non-significance in the latter days: 0.5 comprised in the range of tAUC at days 45 and 50. The overall discriminatory capabilities are fair: C-index of 0.68 ± 0.01. Calibration of the model is fair, with 7 out of 10 points including the perfect calibration curve within their range.

The DCA and MCT at day 60 of the second nomogram are available in Figure 7. The decision curve analysis shows that the nomogram is beneficial compared to systemic policies (“Euthanize none” and “Euthanize all”) in the less prudent ranges, but above 80% a “Euthanize none” policy is preferable. In other words, all users with a decision threshold of 80% chances or less of non-survival would benefit from the model. More prudent users, who would euthanize animals with minimally 80% chances of non-survival, would not benefit from the usage of this model and would prefer never to euthanize an animal regardless of its prognosis.

As described for the first nomogram, the MCT shows an optimum (a point where the cost is minimal) between the two extreme policies (“Euthanize none” and “Euthanize all”). This optimum varies between a probability of non-survival above which euthanasia is decided of 55% (MCT curve for a ratio of 1) and 85% (MCT curve for a ratio of 0.2).

#### 3.2.3. Veterinarian Prognosis

The LogRank test of the veterinarian “best-guess” prognosis was significant (*p* < 0.001). It shows a statistically significant ordered difference of survival between prognosis groups. This means that veterinarians can guess prognosis better than chance.

The DCA at day 60 for the veterinarian prognosis is presented in Figure 8. There is only a benefit using the “best-guess” prognosis as a decision tool for euthanasia for lower probabilities of death. In other words, users that euthanize easily would profit from usage of this nomogram (here, all those who euthanize animals above 60% predicted probability of non-survival). Otherwise, using the veterinarian prognosis is detrimental compared to a “Euthanize none” policy.

## 4. Discussion

This study evaluated an adaptive elastic-net Cox proportional hazards model for prediction of death and culling in dairy cows with severe clinical mastitis in the 60-day period after a first veterinary evaluation. This statistical method is a robust way of obtaining a nomogram [36].

Based on our results, several risk factors were related to the prognosis of cows affected by severe clinical mastitis. Lactation number, recumbency, status of depression, CRT, ruminal motility rate, dehydration status, L-lactate concentration, hematocrit, band neutrophil count, monocyte count, and milk bacteriology results were all included in our first nomogram. Selection of the lactation number can be explained by the higher probability of culling and tendency of old cows to experience more severe signs during mastitis [37]. Being a downer cow is also reported as carrying an inherent higher risk of death [38]. A status of depression may inform about the general severity of the disease. Increased CRT is usually related to systematic inflammatory response syndrome, dehydration, or distributive shock [39], the exclusion of the highest category (more than 2 s) might be explained by its very small size in our sample (13 cases in total). Elevated L-lactate is a sign of hypoperfusion and dehydration and thus is often related to sepsis and endotoxemia [14]. It is one of the factors affecting our nomogram; it may reflect an increased risk of animals affected by severe mastitis [7,40]. Cardiac troponin was not retained in the final model. Hematocrit is also a sign of dehydration [9], and may reflect this perturbation. An increase in band neutrophils (usually called “left shift”) is an indication of a severe state of inflammation [9]. An increase in monocyte count is an indication of chronic inflammation, suppuration, tissue necrosis, or stress response [9]. These two predictors may reflect the severity of the inflammation and the presence of sepsis. Of all the bacteriology results, only the “Gram positive” category was kept in our final model. This may be explained by the high number of cows with *S. aureus* severe clinical mastitis in our sample (13.7%). This bacterium is related to fatal mastitis (which may explain death of the animal), or with chronic clinical or subclinical mastitis (which may explain culling of the animal). Inclusions of paraclinical predictors slightly improved the predictive abilities of our model. This could be explained by their higher objectivity compared to the clinical-only predictor. The method of statistical regression prevents collinearity, and thus statistical redundancy of the predictors. Some of these predictors partially depend on the same biological process (e.g., hematocrit and L-lactates may be influenced by dehydration). Even if they depend on the same process, they both provide statistical information as they were both included.

Discrimination abilities of our nomograms were assessed with the C-index and the tAUC (Figure 2). Reduction of the discriminatory abilities over time (as shown by tAUC) is explainable by the nature of the non-survival: In the first days, the non-survival is more likely linked with death of the animal, while the later non-survival is linked to culling of the animal. The decision to cull an animal is linked to multiple criteria that may or may not be related to the clinical mastitis case (e.g., lameness, somatic cell count history, milk production, infertility, …), which are not accounted for by our method, and which may be highly variable between farmers [41,42]. A straightforward way to assess the discriminative ability of a survival model is the Harrell’s C-index [17]. Here, we found our nomograms to have a fair ability of discrimination (C-index of 0.71 and 0.68). Overall, the predictive ability of our nomograms was fair to good; improvement of predictive abilities would require a larger sample and new predictors. Further studies are needed to improve these two points.

Calibration of our models, assessed with the calibration plot (Figure 2) was good for the first model. We think that penalized regression was a good way of limiting over-fitting in such a case [19,36] as the calibration curves showed no sign of over-fitting (calibration curves close to perfect calibration). The technique worked well and provided models with a fair to good ability to detect animals that will not survive despite treatment. The final number of predictors for the first model was 12, which is slightly higher than the rule of thumb of 10 events per predictor, but breaking that rule was not important if the calibration of the model was correct [19,36]. The second model had only seven predictors. We chose to limit ourselves to an internal validation via bootstrapping, as dividing a small dataset (like ours) into two (one for training and the other for validation) is not a recommended practice in this situation [17].

While statistically able to properly discriminate an issue, some models were not clinically or economically useful [18,34]. Here, we used two tools to help determine the clinical and economical relevance of our models: DCA and the misclassification cost term (MCT). The DCA is a recent tool, but more and more journal editors endorse this tool [34]. It is a simple way to find the added value of the selection method in a function of the sensibility of a user toward excessive intervention (here, euthanasia of the animal). No exact cut-off point for intervention was defined prior to our study. Yet a range of thresholds of 50% to 90% chances of non-survival was deemed as relevant for euthanasia. For any threshold in the aforementioned range, our first model provided benefits (Figure 3). The MCT was plotted for our nomograms (Figure 4). Misclassification cost term is a tool to find an economically optimal cut-off point [43]. With a ratio of 0.5, the optimal threshold in our two nomograms was 75% risk of non-survival (Figure 4). Using that threshold, 30% of cases would be euthanized, a number which is comparable to the fatality rate observed in our sample.

The “best-guess” prognosis of veterinarians was statistically able to properly discriminate the animals by risk categories (significant LogRank test) but provided a low benefit in the investigated range (Figure 8). Considering the absence of clinical relevance, a “Euthanize none” policy seems to be preferable to a euthanasia decision based on the “best-guess” prognosis. However, our categories were very broad (only three possible categories), and another categorization might prove clinically more relevant.

Some caution must be used for the generalization of our nomograms. Our training population had some specificities: Nearly all cows were Holstein, very few mastitis cases were caused by *Str. uberis* or *Klebsiella infections* and many *E. coli* and *S. aureus* infections were present compared to previous studies [4,7,40]. All the animals were treated, and nearly all of them received fluids, antimicrobials, and anti-inflammatories. A lower level of care might lead to a higher fatality rate. We think that the treatment regimens and level of care was similar between veterinary practices involved in the study, as they communicate regularly about their routines. Nonetheless, clustering was not considered in the study due to the low sample size in four of the five involved practices (36 cases total for four practices).

Finally, the studied population were all cows affected by severe clinical mastitis as seen by a veterinarian. This particularity leads to a dependency on the applicability of our nomograms to the rate at which farmers call their veterinarian for severe clinical mastitis cases. Further studies with a more diverse population of farms and veterinarians would help in refining the prognosis of severe clinical mastitis in dairy cattle.

## 5. Conclusions

An innovative method was used to develop and internally validate nomograms to predict non-survival of dairy cows suffering from severe clinical mastitis in the 60-day period following a first veterinary exam in farm settings. These two nomograms were shown to be clinically and economically relevant. The veterinarian’s ”best-guess” prognosis was, on the other hand, not clinically relevant in identifying animals at high risk of dying or culling. These new tools may help practitioners to identify animals that do not require treatment, but rather a humane euthanasia. The use of the nomograms may be facilitated by the dedicated web apps (https://tlp-umontreal.shinyapps.io/paraclinical_nomogram/ and https://tlp-umontreal.shinyapps.io/clinical_nomogram/, accessed on 29 March 2023).

## Figures and Tables

**Figure 1 vetsci-10-00268-f001:**
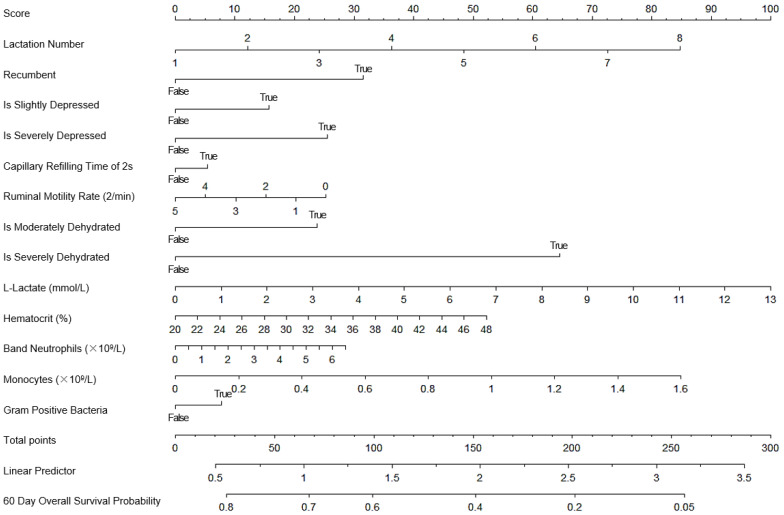
Nomogram with all predictors for predicting 60-day survival of dairy cows coming from 124 herds in Québec, Canada affected by severe clinical mastitis at first veterinary visit.

**Figure 2 vetsci-10-00268-f002:**
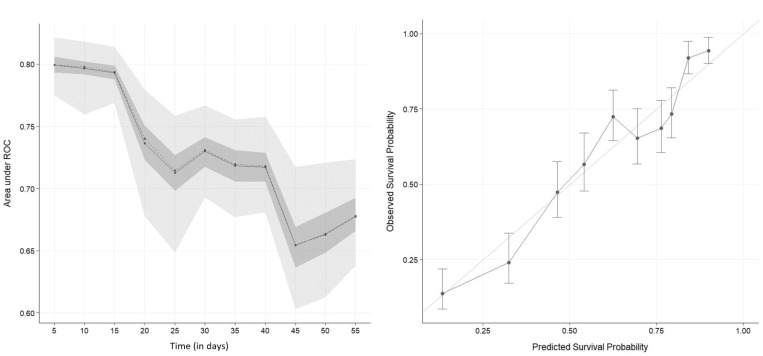
Time-dependent area under the receiver-operator (tAUC) curve (left graph) and calibration curve (right graph) at day 60 of a nomogram predicting survival of dairy cows coming from 124 herds in Québec, Canada with severe clinical mastitis in the 60-day period following a veterinary clinical evaluation. Values were computed based on 100 bootstrap samples. The solid line represents the mean of the tAUC, the dashed line represents the median of the tAUC. The darker interval in the plot shows the 25% and 75% quantiles of the tAUC, the lighter interval shows the minimum and maximum of the tAUC.

**Figure 3 vetsci-10-00268-f003:**
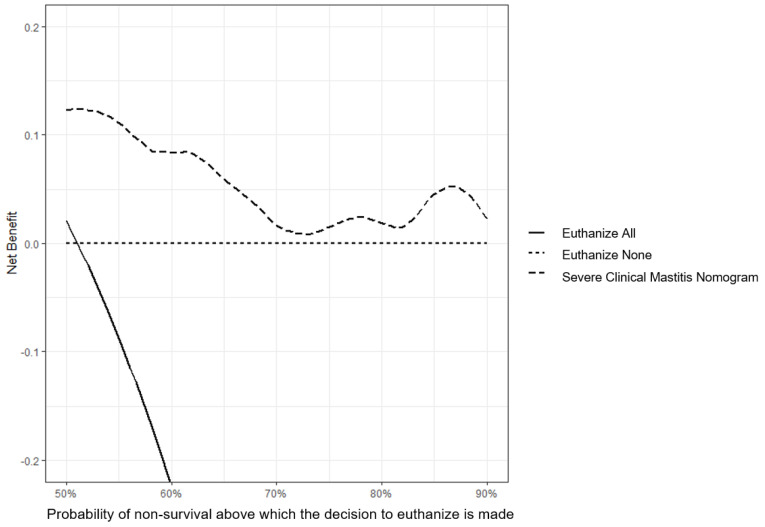
Decision analysis curve at day 60 of a nomogram predicting survival of dairy cows coming from 124 herds in Québec, Canada with severe clinical mastitis in the 60-day period following a veterinary clinical evaluation.

**Figure 4 vetsci-10-00268-f004:**
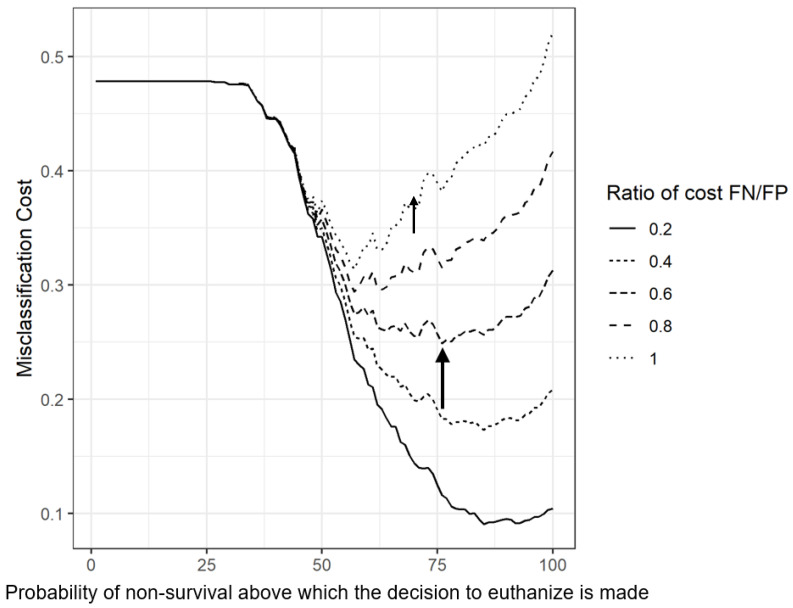
Misclassification cost term at day 60 of a nomogram predicting survival of dairy cows coming from 124 herds in Québec, Canada with severe clinical mastitis in the 60-day period following after first veterinary evaluation (FN: False negative; FP: False Positive). The arrow indicates the minimal cost for a ratio of 0.6, the minimal cost is the economically optimal cut-off point of this ratio.

**Figure 5 vetsci-10-00268-f005:**
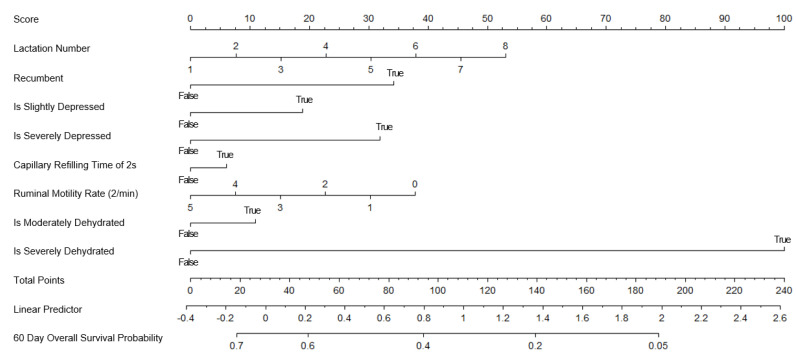
Nomogram predicting the 60-day survival probability of dairy cows coming from 124 herds in Québec, Canada and affected by severe clinical mastitis.

**Figure 6 vetsci-10-00268-f006:**
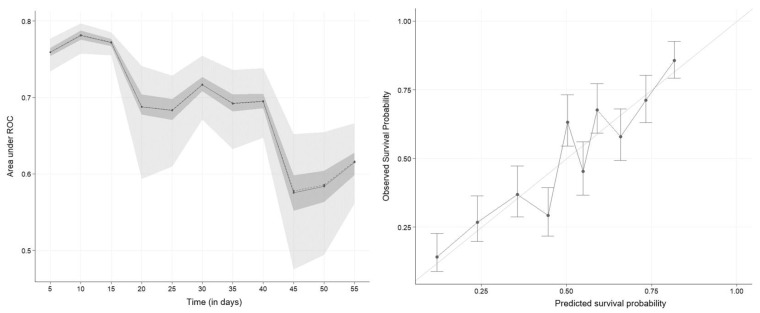
Time-dependent area under the receiver-operator curve (tAUC) (left graph) and calibration curve (right graph) at day 60 of a nomogram built with predictors from the clinical exam, predicting survival of dairy cows coming from 124 herds in Québec, Canada with severe clinical mastitis in function of time after first veterinary clinical evaluation. The values are based on 100 bootstrap samples. The solid line represents the mean of the tAUC, the dashed line represents the median of the tAUC. The darker interval in the plot shows the 25% and 75% quantiles of AUC, the lighter interval shows the minimum and maximum of tAUC.

**Figure 7 vetsci-10-00268-f007:**
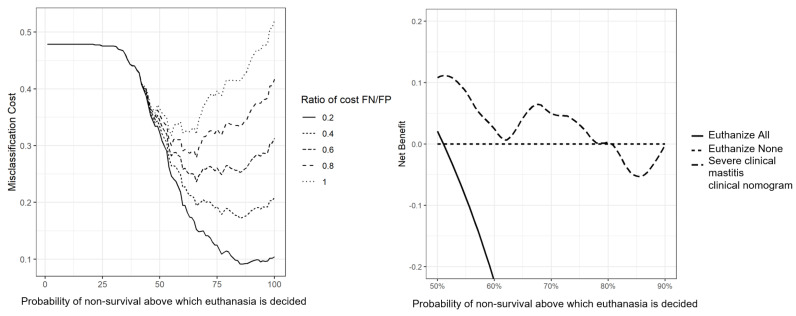
Decision curve analysis (left graph) and misclassification cost term (right graph) of a nomogram built with predictors from the clinical exam, predicting survival of dairy cows coming from 124 herds in Québec, Canada with severe clinical mastitis in the 60-day period following a veterinary clinical evaluation. (FN: False negative; FP: False Positive).

**Figure 8 vetsci-10-00268-f008:**
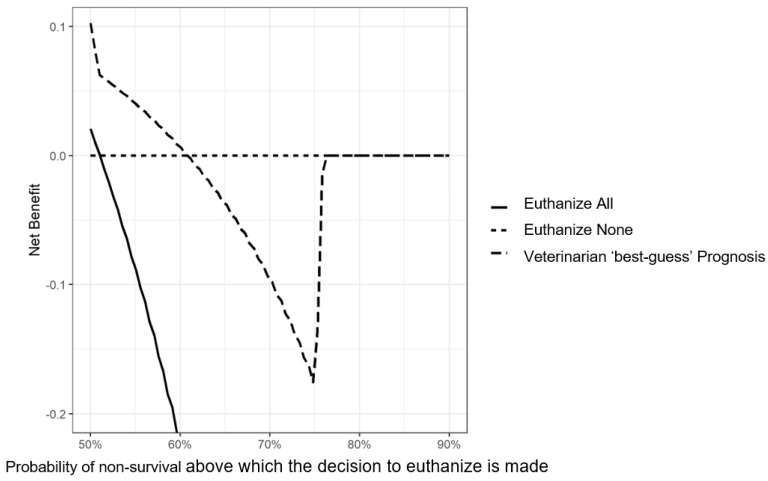
Decision curve analysis of veterinarian “best-guess” prognosis of dairy cows coming from 124 herds in Québec, Canada with severe clinical mastitis in the 60-day period following a veterinary clinical evaluation.

**Table 1 vetsci-10-00268-t001:** Descriptive statistics (overall and stratified by survival status) of clinical examination data from dairy cows (124 herds) affected by severe clinical mastitis in Québec, Canada.

	*N*	Overall *N* = 222 ^1^	Survival *N* = 110 ^1^	Non-Survival *N* = 112 ^1^	*p*-Value ^2^
**Downer cow**	222	34 (15%)	11 (10%)	23 (21%)	0.029
**General state**	215				<0.001
Alert		47 (22%)	33 (31%)	14 (13%)	
Slightly depressed		123 (57%)	58 (55%)	65 (59%)	
Severely depressed		45 (21%)	14 (13%)	31 (28%)	
**Appetite** (compared to normal)	206				0.434
0%		72 (35%)	38 (38%)	34 (32%)	
25%		95 (46%)	46 (46%)	49 (46%)	
50%		39 (19%)	15 (15%)	24 (22%)	
**Temperature** (°C)	219	39.4 (38.7, 40.0)	39.6 (38.7, 40.1)	39.3 (38.6, 39.8)	0.043
**Heart rate** (bpm ^3^)	219	100 (90, 110)	100 (88, 102)	100 (90, 120)	0.004
**Respiratory rate** (mpm ^4^)	208	30 (24, 40)	29 (24, 38)	32 (24, 40)	0.063
**Mucosal aspect**	212				0.267
Normal		140 (66%)	75 (72%)	65 (60%)	
Pale		23 (11%)	8 (7.7%)	15 (14%)	
Congested		49 (23%)	21 (20%)	28 (26%)	
**Capillary refilling time**	222				0.320
<2 s		132 (59%)	71 (65%)	61 (54%)	
2 s		77 (35%)	33 (30%)	44 (39%)	
>2 s		13 (5.9%)	6 (5.5%)	7 (6.2%)	
**Ruminal motility rate** (/2 min)	209	1 (0, 2)	1 (0, 2)	0 (0, 1)	<0.001
**Ruminal motility description**	145				<0.001
Complete		47 (32%)	36 (44%)	11 (17%)	
Incomplete		98 (68%)	45 (56%)	53 (83%)	
**Dehydration**	218				0.002
<5%		44 (20%)	28 (26%)	16 (14%)	
5–7%		148 (68%)	74 (69%)	74 (67%)	
8–10%		22 (10%)	5 (4.7%)	17 (15%)	
10%		4 (1.8%)	0 (0%)	4 (3.6%)	
**Feces aspect**	209				0.016
Normal		1 (0.5%)	1 (1.0%)	0 (0%)	
Blood		37 (18%)	16 (15%)	21 (20%)	
Diarrhea		33 (16%)	12 (11%)	21 (20%)	
Dry		1 (0.5%)	1 (1.0%)	0 (0%)	
Fibrin		2 (1.0%)	1 (1.0%)	1 (1.0%)	
Melena		97 (46%)	60 (57%)	37 (36%)	
Undigested		38 (18%)	14 (13%)	24 (23%)	
**Feces quantity**	214				0.015
Normal		74 (35%)	42 (40%)	32 (29%)	
Low		130 (61%)	61 (59%)	69 (63%)	
None		10 (4.7%)	1 (1.0%)	9 (8.2%)	

^1^ *n* (%); Median (IQR) ^2^ Comparison of survival and non-survival groups, Pearson’s chi-squared test; Wilcoxon rank sum test; Fisher’s exact test ^3^ Bpm: beats per minute ^4^ mpm: movements per minute.

**Table 2 vetsci-10-00268-t002:** Descriptive statistics (overall and stratified by survival status) of complete blood count, cardiac troponin 1, and L-lactate measurement data from dairy cows (124 herds) affected by severe clinical mastitis in Québec, Canada.

	*N*	Overall *N* = 222 ^1^	Survival *N* = 110 ^1^	Non-Survival *N* = 112 ^1^	*p*-Value ^2^
**Troponin** (mmol/L)	219	0.03 (0.00, 0.18)	0.02 (0.00, 0.07)	0.06 (0.01, 0.28)	<0.001
**L-lactate** (mmol/L)	222	1.60 (0.90, 2.70)	1.30 (0.80, 2.28)	1.85 (1.10, 2.92)	0.001
**Erythrocytes** (* 10^12^/L)	189	6.49 (5.75, 7.14)	6.31 (5.51, 6.87)	6.79 (6.11, 7.44)	<0.001
**Hemoglobin** (g/L)	189	114 (102, 126)	108 (99, 119)	121 (108, 130)	<0.001
**Hematocrit** (%)	189	33.0 (29.0, 37.0)	31.0 (29.0, 34.0)	35.0 (31.5, 37.0)	<0.001
**Mean globular volume** (fl)	189	51.0 (49.0, 53.0)	51.0 (48.0, 53.0)	51.0 (49.0, 53.0)	0.481
**Mean corpuscular hemoglobin** (pg)	189	17 (16, 18)	17 (16, 18)	17 (16, 18)	>0.9
**Mean corpuscular** **hemoglobin concentration** (g/L)	189	342 (336, 347)	341 (336, 347)	342 (336, 347)	>0.9
**Platelets** (* 10^9^/L)	188	195 (112, 282)	214 (119, 331)	158 (108, 242)	0.028
**Leucocytes** (* 10^9^/L)	189	4.2 (2.0, 6.9)	3.3 (1.8, 6.5)	4.7 (2.5, 7.7)	0.10
**Neutrophils** (* 10^9^/L)	189	0.97 (0.40, 2.59)	0.85 (0.40, 2.27)	1.21 (0.42, 2.80)	0.662
**Band neutrophils** (* 10^9^/L)	189	0.04 (0.00, 0.32)	0.04 (0.00, 0.32)	0.02 (0.00, 0.32)	0.902
**Lymphocytes** (* 10^9^/L)	189	1.70 (0.97, 2.66)	1.46 (0.92, 2.20)	1.86 (1.04, 3.18)	0.101
**Monocytes** (* 10^9^/L)	189	0.11 (0.05, 0.34)	0.07 (0.02, 0.28)	0.16 (0.07, 0.42)	0.003
**Eosinophils** (* 10^9^/L)	189	0.02 (0.00, 0.14)	0.02 (0.00, 0.14)	0.02 (0.00, 0.14)	0.809
**Basophils** (* 10^9^/L)	189	0.00 (0.00, 0.00)	0.00 (0.00, 0.00)	0.00 (0.00, 0.00)	0.556
**Fibrinogen** (g/L)	188	6 (4, 7)	6 (5, 7)	6 (4, 8)	0.904
**Presence of toxic neutrophils**	190	22 (12%)	9 (9.6%)	13 (14%)	0.486

^1^ Median (IQR); *n* (%) ^2^ Comparison of survival and non-survival groups, Wilcoxon rank sum test; Pearson’s chi-squared test; Fisher’s exact test.

**Table 3 vetsci-10-00268-t003:** Descriptive statistics (overall and stratified by survival status) of milk bacteriology data from dairy cows (124 herds) affected by severe clinical mastitis in Québec, Canada.

	Overall *N* = 222	Survival *N* = 110	Non-Survival *N* = 112
No pathogen	16 (7%)	13 (12%)	3 (3%)
Gram positive bacteria ^1^	43 (19%)	17 (15%)	26 (23%)
Gram negative bacteria ^2^	112 (50%)	57 (52%)	55 (49%)
Mixed infections ^3^	10 (5%)	4 (4%)	6 (5%)
Yeasts	2 (1%)	2 (2%)	0 (0%)
Contaminated	1 (1%)	1 (1%)	0 (0%)
Missing data	38 (17%)	16 (15%)	22 (20%)

^1^. Gram negative: *Escherichia coli* (*n* = 93); *Klebsiella* spp. (*n* = 14); *Pasteurella aeruginosa* (*n* = 2); *Enterobacter* spp. (*n* = 2); *Serratia* spp. (*n* = 1). ^2^. Gram positive: *Staphylococcus aureus* (*n* = 21); *Streptococcus uberis* (*n* = 4); *Bacillus* spp. (*n* = 5); *Staphylococcus* spp. (*n* = 3); *Streptococcus dysgalactiae* (*n* = 5); *Trueperella pyogenes* (*n* = 5). ^3^. Mixed infections: *Escherichia coli* and *Staphylococcus aureus* (*n* = 4); *Escherichia coli* and *Staphylococcus* spp. (*n* = 2); *Klebsiella* spp. and *Staphylococcus* spp. (*n* = 1); *Serratia* spp. and *Staphylococcus* spp. (*n* = 1); *Streptococcus dysgalactiae* and *Trueperella pyogenes* (*n* = 2).

## Data Availability

The data presented in this study are available on request from the corresponding author.

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
