# Peer review of "Development of a Nomogram to Estimate the 60-Day Probability of Death or Culling Due to Severe Clinical Mastitis in Dairy Cows at First Veterinary Clinical Evaluation"

_vetsci, 2023, doi:10.3390/vetsci10040268_

Round 1

Reviewer 1 Report

Development of a nomogram to estimate the 60-day probability of death or culling due to severe clinical mastitis in dairy cows at first veterinary clinical evaluation

The article is very interesting, with an innovative approach to predict the non-survival of dairy cows suffering from severe clinical mastitis in the 60 days after the first veterinary exam, which has practical value for early euthanasia decisions. This aspect has several implications, such as avoiding unnecessary animal suffering, better use of labour, less psychological suffering for the farmer, and cost reduction. Also, the use of nomograms is greatly facilitated by the proposed web apps.

The article is well organized and clearly written, and as such, it is simple to follow. Despite this, some improvements can be made to make the text clearer. The text approaches the subject with scientific correctness. The Tables and Figures are relevant for understanding the article. For Figures, it would be necessary to improve quality and uniformity. The material and methods are clearly described, allowing for a perfect understanding by other researchers. The results are well discussed with the existing knowledge on the subject. Finally, the results support the conclusions indicating the value of the nomogram approaches was clinically and economically relevant in the early decision to identify animals at high risk of dying or culling.

Some detailed comments are below:

L28 showed a good calibration and ability of discrimination. “change with” showed a good calibration and ability to discriminate.

L42 please consider changing "are associated" with "are related to". The word associated repeatedly appears in this text.

L49 It would be useful for veterinarians and dairy producers to be able to properly estimate the survival probability of this condition. “change with” It would be helpful for veterinarians and dairy producers to estimate this condition's survival probability properly.

L300 Please delete “are presented in Error! Reference source not found.

Table 1 please maintain p-value decimals uniform.

Figure 2 Please check the figures and improve their quality.

Please Place the caption after the Figure. Check for all figures. 

Figure 3 please improve image quality. Attention to the figure legend

Although the article already has many figures, the authors should consider introducing figures S3 to S5 in the text. It makes understanding the text more straightforward.

L428 it may reflect increased risk of animals affected by a severe mastitis “change with” it may reflect an increased risk of animals affected by severe mastitis

L460 The second model had seven predictors only. “change with” The second model had only seven predictors.

L495 dependency of the applicability “change with” dependency on the applicability

Reviewer 2 Report

Major comments

A general comment is about the long follow up after recruitment. The prediction is clearly better during the first 10 days. The longer the duration, the higher the probability that the cow is culled or euthanized for another reason than the mastitis itself. Indeed, the negative outcome could be completely independent of the mastitis event, and as such, not predicted by the clinical and paraclinical variables. It would have been informative to describe the cause of the negative outcome (death or early culling).

The paragraph on the misclassification cost must be developed to make it clearer. It should also be noted that all cows are not equal. The economic impact is not uniform and depends on the time spent after the mastitis event and the reason for culling: poorly productive cow due to mammary inflammation and fibrosis (with Gram-negative bacteria) or a cow with poor milk quality due to a high somatic cell count and the absence of cure with Gram-positive bacteria like Staphylococcus aureus do not contribute equally to the revenue.

The features of the second nomogram with clinical indicators only is not that different from the first one. The authors did not discuss much the utility of paraclinical variables to the decision. In reality, a small number of paraclinical variables may help, and some are apparently redundant as they probably provide very similar information. The practical consequences must be discussed further how to improve decision for euthanasia and which paraclinical indicators could help in that decision.

A weakness of the study is the absence of an external validation of the models that makes this work still preliminary and raises caution before implementation or generalization to other contexts. The Authors are aware of this limitation and appropriately inform and evaluate these points in the discussion, notably due to a possible difference of etiology, treatment as the main factors.

Minor comments

Line 19: remove a case of
Line 25, 332, 417: being a downer cow could be replaced by recumbency
Line 26, 332: presence of depression could be depression intensity
Line 36: change mammary secretion instead of milk secretion
Line 40: In the first paragraph, the description is related to severe clinical mastitis. If bacteria are prone to produce severe signs, it is not the case for yeast and algae. So change it by mastitis.
Line 49: replace because of its severe impact by because of the negative impact of severe mastitis on cows
Line 129: it would be more precise to indicate hyperthermia rather than fever, as fever recovers different signs including increased temperature, heart and respiratory rates.
Line136: ruminal contraction rate (or better ruminal motility rate) and ruminal contraction frequency are the same. Please choose one term to describe rumen motility
Line 150 and following: please distinguish measured values from calculated ones.
Line 159: a sterile milk sample does not contain bacteria. Please replace by a milk sample collected aseptically or under aseptic conditions
Line 164: For bacteriology results, what is the reason to set a threshold at 100 or more CFU/mL, which is much higher than generally considered for a positive milk sample?
Line 288: three cows were enrolled twice. The probability that the first mastitis event influenced the second one is high as it could be a relapse of the first infection. Why did the Authors keep these cows, that logically would be excluded, except if they have the proof that the events are independent?
Line 300: error in the reference of Table 1
Line 301: Table 1: indicate in the methods how was the appetite estimated by the producer? The rumen fill would be an alternative indicator of appetite decline.
Line 309: non-steroidal anti-inflammatory drugs rather than anti-inflammatories
Line 340: index appears twice
Line 358: figure 2: x-axis legend: time (in days) duplicated
Line 375: above which you decide to euthanize should be above which the decision to euthanize is made

Major comments

A general comment is about the long follow up after recruitment. The prediction is clearly better during the first 10 days. The longer the duration, the higher the probability that the cow is culled or euthanized for another reason than the mastitis itself. Indeed, the negative outcome could be completely independent of the mastitis event, and as such, not predicted by the clinical and paraclinical variables. It would have been informative to describe the cause of the negative outcome (death or early culling).

The paragraph on the misclassification cost must be developed to make it clearer. It should also be noted that all cows are not equal. The economic impact is not uniform and depends on the time spent after the mastitis event and the reason for culling: poorly productive cow due to mammary inflammation and fibrosis (with Gram-negative bacteria) or a cow with poor milk quality due to a high somatic cell count and the absence of cure with Gram-positive bacteria like Staphylococcus aureus do not contribute equally to the revenue.

The features of the second nomogram with clinical indicators only is not that different from the first one. The authors did not discuss much the utility of paraclinical variables to the decision. In reality, a small number of paraclinical variables may help, and some are apparently redundant as they probably provide very similar information. The practical consequences must be discussed further how to improve decision for euthanasia and which paraclinical indicators could help in that decision.

A weakness of the study is the absence of an external validation of the models that makes this work still preliminary and raises caution before implementation or generalization to other contexts. The Authors are aware of this limitation and appropriately inform and evaluate these points in the discussion, notably due to a possible difference of etiology, treatment as the main factors.

Minor comments

Line 19: remove a case of
Line 25, 332, 417: being a downer cow could be replaced by recumbency
Line 26, 332: presence of depression could be depression intensity
Line 36: change mammary secretion instead of milk secretion
Line 40: In the first paragraph, the description is related to severe clinical mastitis. If bacteria are prone to produce severe signs, it is not the case for yeast and algae. So change it by mastitis.
Line 49: replace because of its severe impact by because of the negative impact of severe mastitis on cows
Line 129: it would be more precise to indicate hyperthermia rather than fever, as fever recovers different signs including increased temperature, heart and respiratory rates.
Line136: ruminal contraction rate (or better ruminal motility rate) and ruminal contraction frequency are the same. Please choose one term to describe rumen motility
Line 150 and following: please distinguish measured values from calculated ones.
Line 159: a sterile milk sample does not contain bacteria. Please replace by a milk sample collected aseptically or under aseptic conditions
Line 164: For bacteriology results, what is the reason to set a threshold at 100 or more CFU/mL, which is much higher than generally considered for a positive milk sample?
Line 288: three cows were enrolled twice. The probability that the first mastitis event influenced the second one is high as it could be a relapse of the first infection. Why did the Authors keep these cows, that logically would be excluded, except if they have the proof that the events are independent?
Line 300: error in the reference of Table 1
Line 301: Table 1: indicate in the methods how was the appetite estimated by the producer? The rumen fill would be an alternative indicator of appetite decline.
Line 309: non-steroidal anti-inflammatory drugs rather than anti-inflammatories
Line 340: index appears twice
Line 358: figure 2: x-axis legend: time (in days) duplicated
Line 375: above which you decide to euthanize should be above which the decision to euthanize is made
